

# Upper-room ultraviolet air disinfection might help to reduce COVID-19 transmission in buildings: a feasibility study

Clive B. Beggs[1],[*] and Eldad J. Avital[2],[*]

[1] Carnegie School of Sport, Leeds Beckett University, Leeds, UK
[2] School of Engineering and Materials Science, Queen Mary University of London, London, UK
[*] These authors contributed equally to this work.

## ABSTRACT

As the world's economies come out of the lockdown imposed by the COVID-19 pandemic, there is an urgent need for technologies to mitigate COVID-19 transmission in confined spaces such as buildings. This feasibility study looks at one such technology, upper-room ultraviolet (UV) air disinfection, that can be safely used while humans are present in the room space, and which has already proven its efficacy as an intervention to inhibit the transmission of airborne diseases such as measles and tuberculosis. Using published data from various sources, it is shown that the SARS-CoV-2 virus, the causative agent of COVID-19, is highly likely to be susceptible to UV-C damage when suspended in air, with a UV susceptibility constant likely to be in the region 0.377–0.590 $m^2$/J, similar to that for other aerosolised coronaviruses. As such, the UV-C flux required to disinfect the virus is expected to be acceptable and safe for upper-room applications. Through analysis of expected and worst-case scenarios, the efficacy of the upper-room UV-C approach for reducing COVID-19 transmission in confined spaces (with moderate but sufficient ceiling height) is demonstrated. Furthermore, it is shown that with SARS-CoV-2, it should be possible to achieve high equivalent air change rates using upper-room UV air disinfection, suggesting that the technology might be particularly applicable to poorly ventilated spaces.

# INTRODUCTION

Since the emergence of COVID-19 in January 2020 there has been considerable interest in the use of ultraviolet (UV) light to disinfect blood plasma (*Eickmann et al., 2020*; *Keil et al., 2020*; *Ragan et al., 2020*), equipment (*Card et al., 2020*; *Derraik et al., 2020*; *Hamzavi et al., 2020*; *Heimbuch & Harnish, 2019*) and air (*Morawska et al., 2020*), in the hope that this might reduce transmission of the disease. In particular, upper-room ultraviolet germicidal irradiation (UVGI), a technology that disinfects room air, has

Corresponding author
Eldad J. Avital, e.avital@qmul.ac.uk

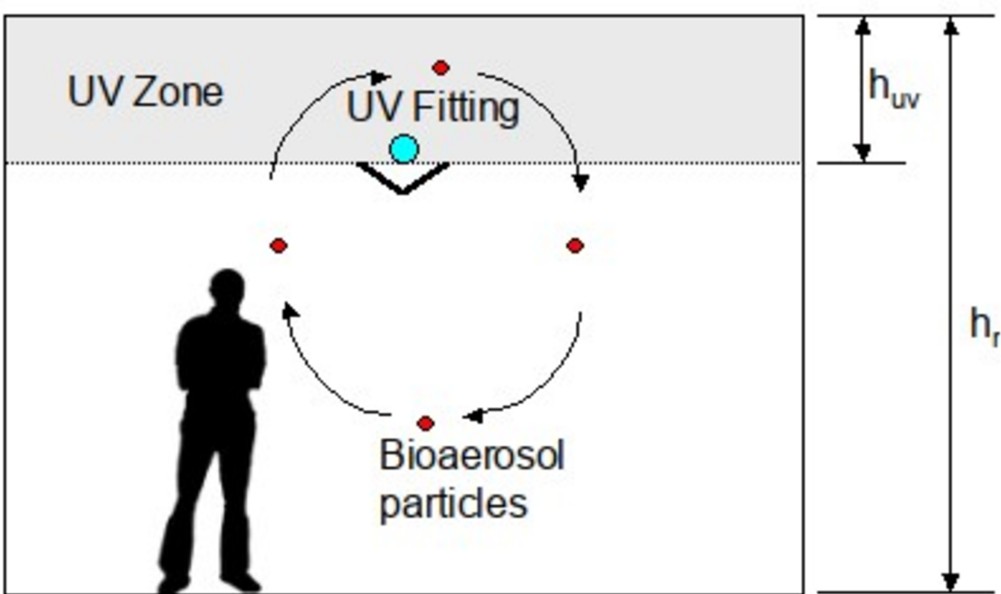

**Figure 1 An upper-room UVGI installation.**

been muted as a potential intervention that might prove effective against COVID-19 (*Morawska et al., 2020*; *Nardell & Nathavitharana, 2020*; *Skorzewska, 2020*). Upper-room UVGI utilizes UV-C light at wavelengths close to 254 nm to create an irradiation field above the heads of room occupants (Fig. 1) that disinfects aerosolised bacteria and viruses suspended in the air (*Beggs et al., 2006*; *Beggs & Sleigh, 2002*; *Noakes, Khan & Gilkeson, 2015*). Because UV-C light is harmful to humans, such systems utilize louvers or shields that obscure the UV lamps from eyesight so that room occupants are kept safe. As such, upper-room UVGI is a well-established technology (*First et al., 1999a*, *1999b*) that has proven effective as a public health intervention to prevent the spread of airborne diseases such as measles (*Nardell & Nathavitharana, 2019*) and tuberculosis (TB) (*Escombe et al., 2009*; *Mphaphlele et al., 2015*) in buildings.

Given that COVID-19 can be transmitted by the inhalation of aerosolised respiratory droplets containing the SARS-CoV-2 virus (*Beggs, 2020*; *Miller et al., 2020*; *Morawska et al., 2020*; *Stadnytskyi et al., 2020*), and that several studies have recovered viral RNA from hospital air samples (*Chia et al., 2020*; *Guo et al., 2020*; *Jiang et al., 2020*; *Santarpia et al., 2020*), there is reason to believe that upper-room UVGI might be effective at "killingx201D; (inactivating) SARS-CoV-2 virions in the air, thus reducing the transmission of COVID-19 in buildings and other enclosed spaces. However, this presupposes that the technology is capable of delivering irradiation doses high enough to inactivate SARS-CoV-2 virions in respiratory droplets suspended in the air, something that has not yet been proven. Given this and the urgent need to develop interventions to break the chain of infection associated with COVID-19, we designed the short feasibility study reported here with the aim of evaluating whether or not upper-room UVGI might be an effective intervention against COVID-19.

## METHODS

### Theory

At any point in time the amount of viral inactivation (disinfection) achieved for a given UV radiant flux (irradiance) can be described using the following first order decay equation (*McDevitt, Rudnick & Radonovich, 2012*).

$$N_t = N_0 \times e^{-Z.E.t} \tag{1}$$

where: $N_0$ and $N_t$ are the number of viable viral particles (virions) at time zero and $t$ seconds respectively; $Z$ is the UV susceptibility constant for the virus (m$^2$/J); $E$ is the radiant (irradiation) flux (W/m$^2$); and $t$ is time in seconds.

The UV irradiation dose received by the virus is simply:

$$H = E \times t \tag{2}$$

where: $H$ is the observed UV irradiation dose (J/m$^2$).

By combining Eqs. (1) and (2), and rearranging we can obtain a value for $Z$.

$$Z = -\frac{1}{H} \times \ln\left(\frac{N_t}{N_0}\right) = -\frac{1}{H} \times \ln(f) \tag{3}$$

where: $f$ is the survival fraction.

Because the relationship between the UV dose and the natural logarithm of the survival fraction is broadly linear for most viral species, it means that the behavior of any given virus exposed to UV-C light can be succinctly described by the value of $Z$, irrespective of the actual UV dose applied. As such, for any given viral species, if the value of $Z$ is known, then it should be possible to predict with reasonable accuracy how the virus will behave when exposed to a given UV-C dose in any context. Microbes that exhibit larger $Z$ values are more susceptible to UV damage, whereas those with small $Z$ values are more difficult to inactivate.

UV inactivation plots for most viral species tend to be straight lines, although some might exhibit a curve (*Kariwa, Fujii & Takashima, 2006*). Notwithstanding this, the model described in Eq. (1) is still a good approximation for most viral species (*McDevitt, Rudnick & Radonovich, 2012*) up until the point where the "target" becomes saturated with UV photons. At this point, because all the virions have already been inactivated, increasing the UV dose further has no effect and so the linear relationship between UV dose and the log reduction becomes decoupled, with the result that the $Z$ value no long applies.

Instead of quantifying UV inactivation in terms of survival fraction, many researchers, particularly those working in biology, describe the reduction in the microbial count in terms of log reduction, which can be converted to survival fraction as follows:

$$f = \frac{1}{10^A} \tag{4}$$

where: $A$ is the $\log_{10}$ reduction in the number of virions.

Specifically, with regard to upper-room UVGI, once the $Z$ value has been obtained for the target microbe, it is then possible to determine the irradiation flux required to disinfect

it, using the methodology described in *Beggs & Sleigh (2002)*. This method makes the assumption that the room air is well mixed, which is a reasonable approximation for most applications (*Beggs & Sleigh, 2002*). If this is the case, then the average particle residence time, $t_{res}$ (in seconds) in the room space will be:

$$t_{res} = \frac{1}{n} \times 3{,}600 \tag{5}$$

where: $n$ is the room ventilation rate in air changes per hour (AC/h).

From Eq. (5) it can be approximated that the average particle residence time in the upper-room UV field, $t_{uv}$ (in seconds) will be:

$$t_{uv} = t_{res} \times \frac{h_{uv}}{h_r} \tag{6}$$

where: $h_r$ is the floor-to-ceiling height (m), and $h_{uv}$ is the depth of the upper-room UV zone (m) (see Fig. 1).

Because $Z$ values are often determined experimentally using microbes suspended in liquids or on surfaces, it may be necessary to adjust the $Z$ value for use with upper-room UVGI systems (*Beggs et al., 2006*; *Yang et al., 2017*), as follows:

$$Z_{ur} = Z \times c_{ur} \tag{7}$$

where: $Z_{ur}$ is the effective upper-room $Z$ value (m$^2$/J), and $c_{ur}$ is a correction coefficient.

For practical purposes, $Z_{ur}$ can be assumed to be the same as the $Z$ value achieved when a given microbe is irradiated in an aerosol.

So if we assume that the air in a room is well mixed, by combining Eqs. (2), (3) and (6) it is possible to compute the average irradiation flux, $E_r$, that is required to achieve a desired survival fraction, $f_r$.

$$E_r = -\frac{1}{(Z_{ur} \times t_{uv})} \times \ln(f_r) \tag{8}$$

Alternatively, the disinfection achieved by an upper-room UVGI system can be thought of as being equivalent to additional air changes in the room space (*McDevitt et al., 2008*). In this scenario, the UV rate constant, $k_{uv}$, which can be thought of as the equivalent air change rate per second, can be determined using (*Beggs et al., 2006*):

$$k_{uv} = Z_{ur} \times E \times \frac{h_{uv}}{h_r} \tag{9}$$

So, in a ventilated room in which contamination ceases at time zero, we can utilize both the UV rate constant, $k_{uv}$, and a rate constant, $k_v$, for the ventilation (i.e., $n \div 3{,}600$), to produce a decay model for the room space.

$$C_t = C_0 \times e^{-(k_v + k_{uv} + k_d)t} \tag{10}$$

where: $C_0$ and $C_t$ are the concentrations of viable viral particles in the room space (virions/m$^3$) at time zero and $t$ seconds respectively; $k_v$ is the ventilation rate constant; $k_d$ is

the particulate deposition rate constant (e.g., 0.0014 s$^{-1}$ (*Stadnytskyi et al., 2020*)); and $t$ is time in seconds.

## Analysis of published data

A search of the relevant scientific literature (i.e., published literature, pre-prints and relevant websites) was undertaken to identify published data relating to the UV irradiation of the three closely related coronaviruses: SARS-CoV-2, the causative agent of COVID-19; SARS-CoV-1, the causative agent of severe acute respiratory syndrome (SARS); and MERS-CoV, the causative agent of middle east respiratory syndrome (MERS).

Because the experimental methods used in the various UV studies varied greatly, as did the level of detail reported, it was necessary to adopt a standardized approach so that valid comparisons could be made. It was therefore decided that, rather than estimating the $Z$ value for a nominal log one reduction (i.e., D$_{90}$) as others have done (*Kowalski, 2010*), we would instead use the log reduction values and UV doses reported in the various studies to calculate the respective $Z$ values using Eq. (3). In so doing, we were able to utilize the results from studies that would otherwise be excluded because the log reductions achieved were far in excess of one. Where researchers performed experiments using a range of UV doses, we calculated the $Z$ value for two UV doses, one near the start of the inactivation process and the other just before the saturation point. So as to avoid bias due to pseudo-replication, when computing the average $Z$ values for the respective viral species, we first aggregated the $Z$ values reported for the various individual studies and then used the aggregated values to calculate the overall mean $Z$ values for the respective viruses.

In order to compare the $Z$ values for the coronaviruses with those for influenza, we utilized experimental results produced by *Heimbuch & Harnish (2019)* who irradiated coupons of respirator material inoculated with SARS-CoV-1 and MERS-CoV, as well as four strains of influenza A, allowing direct comparisons to be made between the viral species.

## Estimating an effective upper-room $Z$ value for aerosolised SARS-CoV-2

In order to evaluate how SARS-CoV-2 might behave in the presence of UV-C when aerosolised, we reviewed the available literature on the subject (*Jensen, 1964*; *Kowalski, 2010*; *Kowalski et al., 2000*; *McDevitt, Rudnick & Radonovich, 2012*; *Walker & Ko, 2007*) with the aim of estimating a value for the coefficient, $c_{ur}$, in Eq. (7), which we then used to estimate the effective upper-room $Z$ value, $Z_{ur}$. In order to reflect the uncertainty associated with this, we compared effective $Z$ values for aerosolised coronaviruses reported in the literature with values obtained for SARS-CoV-2 in liquids to obtain the range of possible values for $c_{ur}$.

## Computation of required upper-room UV irradiation flux

Having estimated the value of $Z_{ur}$ for SARS-CoV-2 from the literature, we then used Eqs. (6) and (8) to estimate the average upper-room irradiation flux that would be required

**Table 1 UV-C doses applied and log reductions achieved in various studies relating to the SARS-CoV-1 and MERS-CoV viruses.**

| Virus | UV wave length (nm) | Medium & context | Irradiance (μW/cm²) | Duration (min & s) | UV Dose (mJ/cm²) | Inactivation (log reduction) | References |
|---|---|---|---|---|---|---|---|
| SARS-CoV-1 | UV-C (nr) | Liquid in well plate | >90 | 15 min | >81 | >log 0.602 | *Duan et al. (2003)* |
| SARS-CoV-1 | 254 | Liquid in well plate | 4,016 | 1 min | 241 | log 1.4* | *Darnell et al. (2004)* |
| SARS-CoV-1 | 254 | Liquid in well plate | 4,016 | 6 min | 1,446 | log 4.5* | *Darnell et al. (2004)* |
| SARS-CoV-1 | 254 | Liguid in well plate | 4,016 | 20 min | 4,819 | log 4.1* | *Darnell & Taylor (2006)* |
| SARS-CoV-1 | UV-C (nr) | Liquid in well plate | 134 | 5 min | 40 | log 3.2* | *Kariwa, Fujii & Takashima (2006)* |
| SARS-CoV-1 | UV-C (nr) | Liquid in well plate | 134 | 15 min | 121 | log 5.325 | *Kariwa, Fujii & Takashima (2006)* |
| SARS-CoV-1 | 254 | Respirator surface | 2,300 | 7.25 min | 1,000 | ≥log 4.81 | *Heimbuch & Harnish (2019)* |
| SARS-CoV-1 | 254 | Platelet concentrates | nr | nr | 50 | log 3.05 | *Eickmann et al. (2020)* |
| SARS-CoV-1 | 254 | Platelet concentrates | nr | nr | 100 | ≥log 3.5 | *Eickmann et al. (2020)* |
| MERS-CoV | 254 | Respirator surface | 2,300 | 7.25 min | 1,000 | ≥log 4.5 | *Heimbuch & Harnish (2019)* |
| MERS-CoV | UV-C (nr) | Droplet on glass slip | nr | 5 min | nr | ≥log 5.91 | *Bedell, Buchaklian & Perlman (2016)* |
| ARS-CoV-2 | 254 | Liguid in well plate | 1,082 | nr | 3.7 | log 3.3 | *Bianco et al. (2020)* |
| SARS-CoV-2 | UV-C (nr) | Inoculated material | nr | 6 s | 5.0 | log 2.0 | *Signify (2020)* |
| SARS-CoV-2 | UV-C (nr) | Inoculated material | nr | 25 s | 22.0 | log 6.0 | *Signify (2020)* |
| SARS-CoV-2 | 280 | Liquid in petri dish | 3,750 | 1 s | 3.75 | log 0.9 | *Inagaki et al. (2020)* |
| SARS-CoV-2 | 280 | Liquid in petri dish | 3,750 | 10 s | 37.5 | log 3.0 | *Inagaki et al. (2020)* |

**Notes:**
* Estimated from plots and data presented in source material.
nr, not reported in source material.

to achieve a 50–90% reduction in aerosolised SARS-CoV-2 virions (through the action of the UV-C alone) in a 4.2 × 4.2 × 2.5 m high room space for a range of ventilation rates. These dimensions were chosen because they are typical for an upper-room UVGI installation in which the lamp height is 2.1 m above the floor (*First et al., 1999b*). In the model we assumed that the air was completely mixed, which meant that according to Eq. (6), aerosol particles would spend on average 16% of their room residency time in the UV zone.

In addition to computing the required UV flux, we also wanted to know how a standard upper-room UV fitting might perform when challenged by SARS-CoV-2. In accordance with the guidelines stated by *First et al. (1999b)*, we assumed that the room contained a single 30 W (input) UV-C fitting capable of delivering an average upper-room flux of 50 μW/cm², and modeled its performance in terms of equivalent ventilation rate using Eq. (9).

## RESULTS

### Analysis of the published literature

The results of the literature search are summarized in Table 1, which shows the UV-C (254 nm) doses applied and log reductions achieved in six studies investigating SARS-CoV-1 (*Darnell et al., 2004*; *Darnell & Taylor, 2006*; *Duan et al., 2003*; *Eickmann et al., 2020*; *Heimbuch & Harnish, 2019*; *Kariwa, Fujii & Takashima, 2006*), two studies investigating MERS-CoV (*Bedell, Buchaklian & Perlman, 2016*; *Heimbuch & Harnish, 2019*), and two studies investigating SARS-CoV-2 (*Bianco et al., 2020*; *Signify, 2020*). Table 1 also includes the results of one study that investigated the impact of

**Table 2 UV-A/B doses applied and log reductions achieved in the various studies relating to the disinfection of SARS-CoV-2 and MERS-CoV in blood products when riboflavin is used.**

| Virus | UV wave length (nm) | Medium & context | Irradiance ($\mu W/cm^2$) | Duration (min) | UV Dose (mJ/mL) | Inactivation (log reduction) | References |
|---|---|---|---|---|---|---|---|
| MERS-CoV | 270–360 | Blood plasma + riboflavin (pooled) | nr | nr | 6,240 | ≥log 4.07 | *Keil, Bowen & Marschner (2020)* |
| MERS-CoV | 270–360 | Blood plasma + riboflavin (single donor) | nr | nr | 6,240 | ≥log 4.42 | *Keil, Bowen & Marschner (2020)* |
| SARS-CoV-2 | 270–360 | Blood plasma + riboflavin | nr | 4 | 1,872 | ≥log 2.61 | *Ragan et al. (2020)* |
| SARS-CoV-2 | 270–360 | Blood plasma + riboflavin | nr | 4 | 3,744 | ≥log 4.72 | *Ragan et al. (2020)* |
| SARS-CoV-2 | 270–360 | Blood plasma + riboflavin | nr | nr | 6,240 | ≥log 3.4 | *Keil et al. (2020)* |
| SARS-CoV-2 | 270–360 | Platelets + riboflavin | nr | nr | 6,240 | ≥log 4.53 | *Keil et al. (2020)* |

**Note:**
nr, not reported in source material.

deep-UV light at 280 nm (i.e., the boundary between UV-B and UV-C) on SARS-CoV-2 (*Inagaki et al., 2020*). In addition, three studies were found that used a combination of UV-A and UV-B light (270–360 nm), together with the photosensitiser, riboflavin, to disinfect SARS-CoV-2 (*Keil et al., 2020; Ragan et al., 2020*) and MERS-CoV (*Keil, Bowen & Marschner, 2020*) in blood products (Table 2). Although these studies did not utilize UV-C light, it was nevertheless decided to report the results of these studies here so that direct comparisons could be made between SARS-CoV-2 and MERS-CoV. The MERS-CoV irradiation study by *Bedell, Buchaklian & Perlman (2016)* is included for completeness, even though the authors did not report the UV dose received by the virus, making it impossible to compute a $Z$ value for this study.

The computed $Z$ values for the respective experiments are shown in Table 3 (UV-C and deep-UV) and Table 4 (UV-A/B plus riboflavin). From these it can be seen that the $Z$ values for the MERS-CoV virus were similar in magnitude to those for both SARS-CoV-1 (UV-C) and SARS-CoV-2 (UV-A/B). With UV-C irradiation the mean $Z$ value for SARS-CoV-1 was 0.00489 (SD = 0.00611) $m^2/J$, whereas that for MERS-CoV was 0.00104 $m^2/J$. Likewise, for UV-A/B plus riboflavin the corresponding $Z$ values were 0.00020 (SD = 0.00009) $m^2/J$ and 0.00016 $m^2/J$ for SARS-CoV-2 and MERS-CoV respectively. However, by comparison SARS-CoV-2 appeared to be more susceptible to UV damage than either SARS-CoV-1 or MERS-CoV when irradiated with UV-C (mean $Z$ = 0.14141 (SD = 0.09045) $m^2/J$) and deep-UV light (mean $Z$ = 0.03684 $m^2/J$).

The calculated $Z$ values for influenza UV-C irradiation experiments undertaken by *Heimbuch & Harnish (2019)* are presented in Table 5. These experiments, which were carried out using inoculated coupons of respirator material, revealed that in this context the $Z$ values for the various influenza A strains were of the same order of magnitude as those for SARS-CoV-1 and MERS-CoV.

## Effective upper-room $Z$ values for aerosolised SARS-CoV-2

A review of the literature revealed that relatively few experimental studies have been performed involving the UV irradiation of aerosolised viruses, with only three undertaken

**Table 3 Calculated Z values for the UV-C irradiation experiments.**

| Virus | UV Dose (mJ/cm²) | Inactivation (log reduction) | UV susceptibility constant, Z (m²/J) | References |
|---|---|---|---|---|
| SARS-CoV-1 | >81 | >log 0.602 | 0.00171 | *Duan et al. (2003)* |
| SARS-CoV-1 | 241 | log 1.4* | 0.00134* | *Darnell et al. (2004)* |
| SARS-CoV-1 | 1,446 | log 4.5* | 0.00072* | *Darnell et al. (2004)* |
| SARS-CoV-1 | 4,819 | log 4.1* | 0.00020* | *Darnell & Taylor (2006)* |
| SARS-CoV-1 | 40 | log 3.2* | 0.01833* | *Kariwa, Fujii & Takashima (2006)* |
| SARS-CoV-1 | 121 | log 5.325 | 0.01017 | *Kariwa, Fujii & Takashima (2006)* |
| SARS-CoV-1 | 1,000 | ≥log 4.81 | 0.00111 | *Heimbuch & Harnish (2019)* |
| SARS-CoV-1 | 50 | log 3.05 | 0.01405 | *Eickmann et al. (2020)* |
| SARS-CoV-1 | 100 | ≥log 3.5 | 0.00806 | *Eickmann et al. (2020)* |
| MERS-CoV | 1,000 | ≥log 4.5 | 0.00104 | *Heimbuch & Harnish (2019)* |
| SARS-CoV-2 | 3.7 | log 3.3 | 0.20536 | *Bianco et al. (2020)* |
| SARS-CoV-2 | 5 | log 2.0 | 0.09210 | *Signify (2020)* |
| SARS-CoV-2 | 22 | log 6.0 | 0.06280 | *Signify (2020)* |
| SARS-CoV-2 | 3.75** | log 0.9 | 0.05526 | *Inagaki et al. (2020)* |
| SARS-CoV-2 | 37.5** | log 3.0 | 0.01842 | *Inagaki et al. (2020)* |

Notes:
 * Estimated from plots and data presented in source material.
 ** Using deep-UV light at 222 nm.

**Table 4 Calculated Z values for the UV-A/B irradiation plus riboflavin experiments.**

| Virus | UV Dose (mJ/mL) | Inactivation (log reduction) | UV susceptibility constant, Z (m²/J) | References |
|---|---|---|---|---|
| MERS-CoV | 6,240 | ≥log 4.07 | 0.00015 | *Keil, Bowen & Marschner (2020)* |
| MERS-CoV | 6,240 | ≥log 4.42 | 0.00016 | *Keil, Bowen & Marschner (2020)* |
| SARS-CoV-2 | 1,872 | ≥log 2.61 | 0.00032 | *Ragan et al. (2020)* |
| SARS-CoV-2 | 3,744 | ≥log 4.72 | 0.00029 | *Ragan et al. (2020)* |
| SARS-CoV-2 | 6,240 | ≥log 3.4 | 0.00013 | *Keil et al. (2020)* |
| SARS-CoV-2 | 6,240 | ≥log 4.53 | 0.00017 | *Keil et al. (2020)* |

on a coronavirus (*Buonanno et al., 2020*; *Walker & Ko, 2007*). A summary of the findings of several key studies are presented in Table 6, which reveals that most viral species appear to be relatively easy to disinfect when suspended in droplets in the air. In particular, aerosolised viruses appear to be more vulnerable to UV damage than when they are suspended in a liquid or on a substrate. For example, for the 24 irradiation experiments involving adenoviruses suspended in liquid, reported by *Kowalski (2010)*, the average Z value was 0.00586 m²/J, which is an order of magnitude less than the values of 0.0546 and 0.0390 m²/J for aerosolised adenoviruses, attributed to *Jensen (1964)* and *Walker & Ko (2007)* respectively. Regarding coronaviruses, *Walker & Ko (2007)* also performed experiments on aerosolised murine (mouse) hepatitis virus (MHV) coronavirus in a single pass test rig. This revealed a Z value of 0.377 ± 0.119 m²/J for this virus.

**Table 5 Calculated $Z$ values for the UV-C irradiation experiments for different strains of influenza A tested by Heimbuch & Harnish.**

| Virus | Medium & context | UV Dose (mJ/cm$^2$) | Inactivation (log reduction) | UV susceptibility constant, $Z$ (m$^2$/J) |
|---|---|---|---|---|
| Influenza A (H1N1) | Respirator surface | 1,000 | ≥log 6.01 | 0.00138 |
| Avian influenza A (H5N1) | Respirator surface | 1,000 | ≥log 4.46 | 0.00103 |
| Influenza A (H7N9), A/Anhui/1/2013 strain | Respirator surface | 1,000 | ≥log 5.15 | 0.00119 |
| Influenza A (H7N9), A/Shanghai/1/2013 | Respirator surface | 1,000 | ≥log 5.31 | 0.00122 |

**Table 6 Summary of reported effective $Z$ values for single-pass UV irradiation experiments performed on aerosolised viruses in air.**

| Researchers | Virus | UV-C wavelength (nm) | Effective $Z$ value (m$^2$/J) | Reporter |
|---|---|---|---|---|
| Jensen (1964) | Adenovirus | 254 | 0.0546 | Kowalski et al. (2000) |
| Jensen (1964) | Coxsackie B-1 | 254 | 0.1108 | Kowalski et al. (2000) |
| Jensen (1964) | Influenza A | 254 | 0.1187 | Kowalski et al. (2000) |
| Jensen (1964) | Sindbis virus | 254 | 0.1040 | Kowalski (2010) |
| Jensen (1964) | Vaccinia virus | 254 | 0.1528 | Kowalski et al. (2000) |
| Walker & Ko (2007) | Adenovirus | 254 | 0.0390 | Walker & Ko (2007) |
| Walker & Ko (2007) | MHV coronavirus | 254 | 0.3770 | Walker & Ko (2007) |
| McDevitt, Rudnick & Radonovich (2012) | Influenza A | 254 | 0.2700 | McDevitt, Rudnick & Radonovich (2012) |
| McDevitt et al. (2007) | Vaccinia virus | 254 | 2.5400 | McDevitt et al. (2007) |
| Buonanno et al. (2020) | Human coronavirus 229E | 222 | 0.4100 | Buonanno et al. (2020) |
| Buonanno et al. (2020) | Human coronavirus OC43 | 222 | 0.5900 | Buonanno et al. (2020) |

Buonanno et al. (2020) also performed irradiation experiments on aerosolised coronaviruses, but using UV light at 222 nm (far-UV) rather than 254 nm. They found the $Z$ values for human coronavirus 229E and human coronavirus OC43 to be 0.410 m$^2$/J and 0.590 m$^2$/J respectively. Collectively, these $Z$ values are an order of magnitude greater than the values obtained for SARS-CoV-2 in liquid, implying that when aerosolised, coronaviruses in general and SARS-CoV-2 in particular, are much easier to disinfect compared with when they are presented in liquids or on surfaces. Although we are comparing different species of coronavirus here, evidence from Bedell, Buchaklian & Perlman (2016), who irradiated MHV coronavirus and MERS-CoV in Petri dishes, suggests that it is nonetheless valid. They found that 5 min exposed to a UV-C light source resulted in a 2.71 log reduction for the MHV coronavirus, whereas the same exposure resulted in a 5.91 log reduction for MERS-CoV. This suggests that MHV coronavirus is actually more resistant to UV damage than MERS-CoV, and as such, supports Walker & Ko (2007) conclusion that coronaviruses are much easier to inactivate in the air compared with on surfaces and in liquids.

**Table 7 Predicted average upper-room UV irradiance fluxes required to achieve 50%, 70% and 90% inactivation for SARS-CoV-2 assuming a range of $Z_{ur}$ values and ventilation rates (Assuming $Z_{ur}$ = 0.377 or 0.0377 m²/J).**

| Ventilation rate (AC/h) | Average particle residence time in UV field. (min) | UV susceptibility constant, $Z_{ur}$ (m²/J) | Average irradiance required for 50% inactivation (µW/cm²) | Average irradiance required for 70% inactivation (µW/cm²) | Average irradiance required for 90% inactivation (µW/cm²) |
|---|---|---|---|---|---|
| 1 | 9.6 | 0.3770 | 0.319 | 0.554 | 1.060 |
| 2 | 4.8 | 0.3770 | 0.638 | 1.109 | 2.121 |
| 4 | 2.4 | 0.3770 | 1.277 | 2.218 | 4.241 |
| 6 | 1.6 | 0.3770 | 1.915 | 3.327 | 6.362 |
| 8 | 1.2 | 0.3770 | 2.554 | 4.436 | 8.482 |
| 1 | 9.6 | 0.0377 | 3.192 | 5.544 | 10.604 |
| 2 | 4.8 | 0.0377 | 6.384 | 11.088 | 21.207 |
| 4 | 2.4 | 0.0377 | 12.768 | 22.177 | 42.414 |
| 6 | 1.6 | 0.0377 | 19.152 | 33.266 | 63.621 |
| 8 | 1.2 | 0.0377 | 25.536 | 44.355 | 84.829 |

Comparing the computed $Z$ values for UV-C irradiation experiments on the MHV coronavirus conducted in air (0.37700 m²/J (*Walker & Ko, 2007*)) with those for the SARS-CoV-2 virus in liquid ranging from 0.06280 m²/J (*Signify, 2020*) to 0.20536 m²/J (*Kariwa, Fujii & Takashima, 2006*), it would appear that irradiating the coronavirus in liquid requires a UV-C dose that is in the region 1.8–6.0 times higher than that required when the virus is suspended in air. From this we estimated that the value of the adjustment coefficient $c_{ur}$ would be in a range 0.167–0.545.

## Upper-room UVGI computation results

Because no UV irradiation experiments have to date been performed on aerosols containing the SARS-CoV-2 virus, it was necessary when undertaking the feasibility study to make assumptions regarding an appropriate value of $Z_{ur}$ to use in the upper-room UVGI analysis. With respect to this, because the published mean $Z$ values for the aerosolised coronaviruses were all in the region 0.377–0.590 m²/J, we felt that an assumed $Z$ value in this range would be indicative of how airborne SARS-CoV-2 might behave in a UV-C field. A decision was therefore made to use Walker and Ko's $Z$ value figure of 0.377 m²/J to evaluate the expected performance of the upper-room UVGI installation, because this was considered a conservative value. In addition, because of the uncertainty associated with this assumed value, we introduced a "factor of safety" into our analysis by also modeling a worst-case scenario in which $Z_{ur}$ was 0.0377 m²/J.

Table 7 presents the results of the room analysis using these two values for $Z_{ur}$, for a range of ventilation rates. From this it can be seen that there is a direct inverse relationship between particle residence time in the UV field, $t_{uv}$, and the required irradiation flux, $E_r$, as predicted by Eq. (8). This means that for any given $Z$ value, the value of $E_r$ will double as the room ventilation rate doubles. The table also reveals that there is a direct inverse relationship between $Z_{ur}$ and $E_r$. From the calculated values in this table it can be seen

that if $Z_{ur}$ = 0.377 m$^2$/J, then with an average UV flux of just 10 μW/cm$^2$ it should be possible to achieve >90% inactivation of SARS-CoV-2, even at a ventilation rate of 8 AC/h. However, if in reality, $Z_{ur}$, is 0.0377 m$^2$/J, then all the calculated fluxes would have to increase by a factor of ten to achieve the same results. Given that accepted guidelines (*First et al., 1999b*) recommend for a room 2.5 m high, one 30 W (input) UV lamp per 18.58 m$^2$ of floor area, which will produce an average flux in the region 50 μW/cm$^2$, this means that even under this worst-case scenario it should still be possible to achieve disinfection rates >90% for all but the highest ventilation rates.

When we fixed the UV flux at an average of 50 μW/cm$^2$, we found that for $Z_{ur}$ = 0.377 m$^2$/J the upper-room UVGI installation produced an equivalent air change rate of 108.6 AC/h, whereas if $Z_{ur}$ = 0.0377 m$^2$/J this fell to 10.9 AC/h. These values were constant and unaffected by the actual room ventilation rate.

## DISCUSSION

Analysis of the literature relating to the UV irradiation of coronaviruses clearly reveals that SARS-CoV-2, when in a liquid assay, is relatively easily inactivated by UV light at both 254 nm (*Bianco et al., 2020*; *Signify, 2020*) and 280 nm (*Inagaki et al., 2020*). Indeed, the results in Table 3 suggest that the virus is likely to be more susceptible to UV-C damage than either SARS-CoV-1 or MERS-CoV. Furthermore, the results of the experiments were SARS-CoV-2 was exposed to UV-A/B and riboflavin, suggest that the virus is susceptible to damage, albeit to a lesser extent, caused by UV light at other wavelengths. As such, this appears to support the finding of *Sagripanti & Lytle (2020)* that SARS-CoV-2 is vulnerable to sunlight.

One problem frequently encountered when comparing UV irradiation results from disparate researchers is that experimenters often utilize different methodologies to evaluate log reductions in microbial species, with varying doses of UV administered. In particular, the type of substrate or media used can greatly influence the outcome of the experiment. This is because the substrate or media can absorb the UV photons and shield the virus. Given this, it is important to compare like with like, if this is possible. For this reason we included the results of *Heimbuch & Harnish (2019)* in Tables 3 and 5, because they performed the same irradiation experiment on SARS-CoV-1 and MERS-CoV, as well as on four strains of influenza A, thus allowing direct comparisons to be made. From Tables 3 and 5 it can be seen that the *Z* values for the influenza strains are of a similar order of magnitude as those for the coronaviruses, implying that in this context SARS-CoV-1 and MERS-CoV were about as difficult to inactivate as influenza A. This is a surprising finding, because others have suggested that the UV dose required to inactivate SARS-CoV-2 might be lower than that required to disinfect influenza A (*Sagripanti & Lytle, 2020*). This is because coronaviruses have genomes that are approximately twice as long as that of influenza A, making them in theory much more vulnerable to damage from UV-C (*Sagripanti & Lytle, 2020*). Indeed, in a summary collated from hundreds of published studies by *Kowalski (2010)*, the Z values for influenza A in water were reported as being in the range 0.04800–0.13810 m$^2$/J, much higher than the values achieved by *Heimbuch & Harnish (2019)*. As such, this suggests that the substrate or medium in which

microbes are irradiated plays an important role in influencing the magnitude of the $Z$ value achieved. Indeed, it is well known in other contexts that UV-C light can be attenuated as it passes through liquids (*Mamane, 2008*). When UV light passes through a suspension of particles in water, its intensity is reduced due to both scattering and absorption of the light (*Gregory, 2005*). Absorption occurs because the light beam interacts with atoms and molecules in the liquid to raise their energy level, with the result that energy is lost from the beam, whereas scattering occurs when particulates in the fluid interfere with the UV light making it more diffuse (*Mamane, 2008*). Particulates can also shield microbes from UV light. This means that UV inactivation of microbial suspensions in liquid films >1.2 mm can be greatly inhibited, due to the low penetration depth of UV light through concentrated suspensions (*Cheng et al., 2020*). Consequently, when interpreting the $Z$ values for SARS-CoV-1, SARS-CoV-2 and MERS-CoV in Table 3, it is important to view them as being strictly contextual.

With regard to UV irradiation of aerosolised viruses, very few published experimental studies exist, with only three specifically relating to coronaviruses (*Buonanno et al., 2020*; *Walker & Ko, 2007*). As a result there is a paucity of good quality data relating to UV-C irradiation of SARS-CoV-1, SARS-CoV-2 and MERS-CoV in the air. Consequently, we had to establish whether or not *Walker & Ko (2007)* published $Z$ value of 0.377 m$^2$/J was valid for SARS-CoV-2 in air. Comparison with the Z values presented in Table 3 reveals that this value is considerably greater in magnitude than those achieved for the coronaviruses when they were irradiated in liquid or on equipment substrates. This however, is to be expected given that liquids attenuate UV penetration (*Mamane, 2008*). Also the finding appears to be broadly in keeping with the behavior of adenoviruses when irradiated in air and in liquid. Furthermore, because *Bedell, Buchaklian & Perlman (2016)* found MERS-CoV to be more susceptible to UV-C damage than MHV coronavirus, this strongly supports the use of *Walker & Ko (2007)* Z value for MHV coronavirus as a valid surrogate for SARS-CoV-2 in air. Having said this, because the UV susceptibility of the target microbe is crucial to the performance of any upper-room UVGI installation, our use of Walker and Ko's Z value for the MHV coronavirus to represent SARS-CoV-2 should be treated with caution. For this reason, when we assessed the performance of the upper-room UVGI in our hypothetical room, we used both 0.377 and 0.0377 m$^2$/J in our simulations. In so doing, we effectively modeled both the expected and worst-case scenarios.

The results for the expected and worst-case scenarios in Table 7, strongly suggest that upper-room UVGI, if applied correctly, should be effective at disinfecting SARS-CoV-2 virions suspended in respiratory droplets in the air. This finding is of course very much dependent on the surrogate $Z_{ur}$ value being truly representative for SARS-CoV-2. With respect to this, one limitation of our study is that we did not distinguish between the $Z$ values achieved using a single-pass test rig, such as that used by *Walker & Ko (2007)*, and those achieved in real-life by an upper-room UVGI system. With the latter, because the irradiation process is fragmented, compared with a single-pass system, it is thought that higher UV doses might be required to achieve equivalent levels of inactivation (*Beggs et al., 2006*; *Yang et al., 2017*). However, while this specifically applies to aerosolised

bacteria that can rapidly repair UV damage when the irradiation process becomes fragmented (*Beggs, 2002*), it is not known to what extent this applies to viruses, which are not metabolically active, although it is known that through photoreactivation viruses can repair UV damage (*Weinbauer et al., 1997*). Notwithstanding this, because the *Z* values achieved for coronaviruses irradiated in air (*Buonanno et al., 2020*; *Walker & Ko, 2007*) are very similar in magnitude to those exhibited by *Mycobacterium tuberculosis* (the causative agent of TB) in air (i.e., 0.33–0.48 m$^2$/J (*Riley, Knight & Middlebrook, 1976*)), there is good reason to believe that upper-room UVGI might be effective at mitigating the spread of COVID-19 indoors.

Upper-room UVGI air disinfection is highly dependent on good air mixing occurring between the upper and lower portions of the room space (*Beggs & Sleigh, 2002*; *Nicas & Miller, 1999*; *Noakes, Beggs & Sleigh, 2004*; *Zhu et al., 2013*). In the study presented here we assumed that complete mixing occurred, which although a reasonable approximation in many instances, is not always the case because short circuiting can occur (*Beggs & Sleigh, 2002*). If the room air mixing factor is low, say for example due stratification in a poorly ventilated space, then this can greatly impair the disinfection performance of an upper-room UVGI system (*Beggs & Sleigh, 2002*; *Noakes, Beggs & Sleigh, 2004*). It is therefore important when designing such systems to carefully consider the air movement in the room space, in order to eliminate stagnant regions and maximize air movement through the UV field. In the context of COVID-19, this is particularly important because, unlike TB which is spread via the inhalation of droplet nuclei <5 µm in diameter, it is thought that COVID-19 can be transmitted through the exhalation of larger respiratory droplets <100 µm, which rapidly reduce in size due to evaporation (*Beggs, 2020*; *Liu et al., 2017*; *Xie et al., 2007*) to become aerosols, say, <50 µm in diameter (*Nicas, Nazaroff & Hubbard, 2005*). These larger aerosol particles have settling velocities <0.1 m/s and as such can readily be transported on convective room air currents, with the result that they can remain suspended in room air for many minutes. However, if the velocities of the convection currents drop, then some of the larger aerosol particles may decouple from the air stream and settle out due to gravitational deposition, potentially passing through the breathing zone where they can be inhaled by the room occupants. This is particularly the case if the air is poorly mixed and stagnant regions exist within the room space. Under such circumstances larger aerosol particles might be inhaled without being fully irradiated by the upper-room UV field, undermining the effectiveness of the whole UVGI installation. Consequently, if upper-room UVGI is to be effective against COVID-19, it is important to both promote good room air mixing and also ensure that larger aerosol particles (e.g., <50 µm in diameter) receive the required UV irradiation dose. As such, this may require upper-room UVGI systems to be supplemented with ceiling mounted fans (*Zhu et al., 2013*) or other devices to promote the necessary air movement to ensure that larger aerosol particles are adequately irradiated.

One major advantage of upper-room UVGI is that it can be retrospectively fitted into buildings provided that the floor to ceiling height is large enough to ensure that the UV field does not impinge on room occupants (*First et al., 1999b*). By installing such a system it is possible to effectively "turbo-charge" the efficacy of the ventilation system.

Indeed, in keeping with the findings of *McDevitt et al. (2008)*, our analysis suggests that it is possible to achieve >100 equivalent AC/h by installing upper-room UVGI. Using Eq. (9), we can calculate the UV rate constant, $k_{uv}$, which can be thought of as the equivalent air change rate per second. Once known, this in turn can be used, together with the ventilation and particulate deposition rate constants, $k_v$, and $k_d$, in Eq. (10), to compute the concentration of viral partials in the room space at any point in time.

While our analysis has been able to show that upper-room UVGI has the potential to disinfect the SARS-CoV-2 virus when suspended in room air, we are nonetheless conscious of the limitations of the feasibility study. Chief among these is the fact that we had to assume the value of the upper-room UV susceptibility constant, $Z_{ur}$, for SARS-CoV-2. Although the true value of this constant is likely to be similar to that exhibited by other aerosolised coronaviruses, we cannot know its exact magnitude without further experimental work. Consequently, the results of the study should be considered as indicative only. Also, in the study it was assumed that the room air is well mixed, which, as discussed above, may not necessarily be the case in some applications. In particular, because the model used was relatively simple, we were not able to assess how upper-room UVGI might perform in situations where aerosol particles decouple from the air stream due to gravitational deposition, or remain suspended in the breath zone. It is therefore recommended that future studies investigating the use of upper-room UVGI to prevent COVID-19 transmission use computational fluid dynamics (CFD) to assess the limitations of the technology with respect to the disinfection of larger aerosol particles that might decouple from room air convection currents.

## CONCLUSIONS

In conclusion, we have been able to demonstrate that the SARS-CoV-2 virus is relatively easily inactivated by UV-C light and that when aerosolised the virus is likely to have a UV susceptibility constant, $Z_{ur}$, that is similar to that exhibited by other coronaviruses in air. This suggests that SARS-CoV-2 when suspended in air should be reasonably easy to inactivate using UV light at 254 nm. As such, upper-room UVGI may have potential as an intervention to inhibit the transmission of COVID-19 in buildings, especially in situations where achieving high ventilation rates might otherwise be impractical.

### Funding
The authors received no funding for this work.

### Competing Interests
The authors declare that they have no competing interests.

### Author Contributions
- Clive B. Beggs conceived and designed the experiments, performed the experiments, analyzed the data, prepared figures and/or tables, authored or reviewed drafts of the paper, and approved the final draft.

- Eldad J. Avital conceived and designed the experiments, performed the experiments, analyzed the data, prepared figures and/or tables, authored or reviewed drafts of the paper, and approved the final draft.

## Data Availability

Data is available in Tables 1–7.

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
