# Peer review of "Upper-room ultraviolet air disinfection might help to reduce COVID-19 transmission in buildings: a feasibility study"

_PeerJ, doi:10.7717/peerj.10196_

## Round 0.1 · original submission · Major Revisions

Please address all the comments by the reviewers, but focus specifically on (1) reporting the inactivation” z” values of airborne SARS-CoV-2 when exposed to UV-C light, (2) inability of 1 reviewer to replicate the results from the formula [8], using the data given in Table 7, (3) virus transmitted via droplets which may not be carried to the upper room zone by the indoor airflow, and (4) expanded discussion of the limitations of the well-mixed assumption.

Reviewer 1 ·

Basic reporting

The authors performed a literature review and summarized the reported inactivation “Z’ values of UVC light from different published experiments based on different pathogens in air and surface samples.

The language is clear and the paper is well-written in Engish

Experimental design

Nevertheless, for the pathogen that causes COVID-19, tables 2 and 4 only provide data on UVA or UVB light in a Blood plasma sample from other published studies. The authors did not report the inactivation” z” values of airborne SARS-CoV-2 when exposed to UV-C light. Original or new experimental findings are expected. The simulation employed is a simple tool based on a well-mixed model. The equations are previously established and published in previous research.

Validity of the findings

The z values used in the model are based on assumption and were NOT validated with real experimental data.

Additional comments

see above

·

Basic reporting

The reviewer considered the second paragraph of the abstract to be ”dense” and a little complex to read. If it is within the word count restriction and does not lead to lose of clarity of the main message, some further revision is suggested.

Page 10, Section 4 Discussion – second line “highly lightly” – should be “highly likely”

No further comments

Experimental design

no comment

Validity of the findings

Page 8, Section 3.1 Analysis – second paragraph “mean Z value for SARS-CoV-1 was 0.00489” – however, reviewer found the mean to be 0.00619, likely to be an interpreted or weighted mean? Please check with Authors, and ensure this is clear within the results.

Page 9, Section 3.3 Upper-room results – second paragraph. The main findings seem perfectly acceptable and indeed translate well from the table, to the results, discussion, conclusions and finally to the abstract. However, given this is the basis of the paper, the reviewer decided to try and replicate the results from the formula [8], using the data given in Table 7, as might reader of the paper. This proved difficult, and could be due to reviewer not fully understanding the maths, or a translation of units. Again as above, it is advised to check with Authors.

No further comments

Additional comments

As the reviewer I am satisfied that the Authors have considered the how SARS-CoV-2 might be susceptible to UV-C light by analysing other viruses, trying to develop a UV-C susceptibility constant, Z. Then using this knowledge to investigate the efficacy of an upper-room reduction of the COVID-19 air transmission, and using the results generated to support their conclusions.

·

Basic reporting

Typo in first sentence of Discussion. "Lightly" should be "likely".
Typo in first line of page 12 in Discussion. "This finding is of course is".

Experimental design

No comment

Validity of the findings

Covid-19 can be transmitted via droplets which may not be carried to the upper room zone by the indoor airflow. These larger droplets will eventually either deposit on surfaces or reduce in size due to evaporation, however there is a possibility that they could remain in the breathing zone for a significant period of time. Could the authors comment on this and its implications for the effectiveness of upper room UV?

The analysis assumes a well mixed room. The authors acknowledge this and refer to a paper which illustrates that this is likely an acceptable assumption for most rooms (reference 11). However an expanded discussion of the limitations of this assumption would be of value. Are there occurrences when this may not be a valid assumption and therefore the effectiveness of upper room UV might be reduced?

Additional comments

This is a very timely article by the authors as various control measures to mitigate the transmission rate of Covid-19 indoors are being considered. The authors make a convincing case that aerosolised SARS-CoV-2 is likely to be susceptible to UV-C irradiation using the evidence currently available in the literature. Their analysis demonstrates that high equivalent AC/H could be achieved by installing upper room UV.

---

## Round 0.2 · accepted · Accept

Thank you for your efforts in revising your manuscript in response to reviewer comments.

·

Basic reporting

Overall the further work, with the clearer and improved abstract has enhanced the overall manuscript. Emphasising that the work is a feasibility study is also important, yet this does not detract from it being a timely piece of research.

I did not find any further typographical errors.

No further comments.

Experimental design

No further comments.

Validity of the findings

My previous comments on the validity of the analysis and results have been addressed by the authors to my satisfaction. Having worked through the example given, I am clear the calculations represent the results given.

No further comments.

Additional comments

As the reviewer I am satisfied that the Authors have considered the how SAR-CoV-2 might be susceptible to UV-C light by analysing other viruses, developing a UV-C susceptibility constant, Z. Then using this knowledge to investigate the efficacy of an upper-room reduction of the COVID-19 air transmission, and using the results generated to support their conclusions.

This was as per my previous review, I am now further convinced that this technology is important. Thus this feasibility study is timely and should act as a prompt for a study into how SAR-CoV-2 as an aerosol might be susceptible to UV-C light.

·

Basic reporting

NA

Experimental design

NA

Validity of the findings

NA

Additional comments

I'd like to thank the authors for their response to my comments and am satisfied that they have been adequately addressed.